# A systematic review of dengue outbreak prediction models: Current scenario and future directions

**Xing Yu Leung[1], Rakibul M. Islam[1], Mohammadmehdi Adhami[1], Dragan Ilic[1], Lara McDonald[1], Shanika Palawaththa[1], Basia Diug[1], Saif U. Munshi[2], Md Nazmul Karim[1]***

**1** School of Public Health and Preventive Medicine, Monash University, Melbourne, Victoria, Australia,
**2** Department of Virology, Bangabandhu Sheikh Mujib Medical University, Dhaka, Bangladesh

* nazmul.karim@monash.edu

**Data Availability Statement:** All relevant data are within the manuscript and its Supporting Information files.

## Abstract

Dengue is among the fastest-spreading vector-borne infectious disease, with outbreaks often overwhelm the health system and result in huge morbidity and mortality in its endemic populations in the absence of an efficient warning system. A large number of prediction models are currently in use globally. As such, this study aimed to systematically review the published literature that used quantitative models to predict dengue outbreaks and provide insights about the current practices. A systematic search was undertaken, using the Ovid MEDLINE, EMBASE, Scopus and Web of Science databases for published citations, without time or geographical restrictions. Study selection, data extraction and management process were devised in accordance with the 'Checklist for Critical Appraisal and Data Extraction for Systematic Reviews of Prediction Modelling Studies' ('CHARMS') framework. A total of 99 models were included in the review from 64 studies. Most models sourced climate (94.7%) and climate change (77.8%) data from agency reports and only 59.6% of the models adjusted for reporting time lag. All included models used climate predictors; 70.7% of them were built with only climate factors. Climate factors were used in combination with climate change factors (13.4%), both climate change and demographic factors (3.1%), vector factors (6.3%), and demographic factors (5.2%). Machine learning techniques were used for 39.4% of the models. Of these, random forest (15.4%), neural networks (23.1%) and ensemble models (10.3%) were notable. Among the statistical (60.6%) models, linear regression (18.3%), Poisson regression (18.3%), generalized additive models (16.7%) and time series/autoregressive models (26.7%) were notable. Around 20.2% of the models reported no validation at all and only 5.2% reported external validation. The reporting of methodology and model performance measures were inadequate in many of the existing prediction models. This review collates plausible predictors and methodological approaches, which will contribute to robust modelling in diverse settings and populations.

**Funding:** The author(s) received no specific funding for this work.

**Competing interests:** The authors declare that they have no conflicts of interest.

## Author summary

Dengue is considered as a major public health challenge and a life-threatening disease affecting people worldwide. Over the past decades, numerous forecast models have been developed to predict dengue incidence using various factors based on different geographical locations. Dengue transmission appears to be highly sensitive to climate variability and change, however quantitative models used to assess the relationship between climate change and dengue often differ due to their distribution assumptions, the nature of the relationship and the spatial and/or temporal dynamics of the response. We performed a systematic review to examine current literature surrounding existing quantitative models based on development methodology, predictor variable used and model performance. Our analysis demonstrates several shortcomings in current modelling practice, and advocates for the use of real time primary predictor data, the incorporation of non-climatic parameters as predictors and more comprehensive reporting of model development techniques and validation.

This review collates methodological approaches adopted in the modelling practices in the field across current literature. This will provide an evidence-based framework for upgrading future modelling practice to develop more accurate predictive models with robust techniques. In turn, this also provided an opportunity for the effective distribution of limited public health resources to prepare for demand.

## Introduction

Dengue fever is one of the fastest-spreading mosquitos-borne disease primarily of tropical and subtropical regions and is caused by various dengue virus strains [1,2]. In 2017 alone, over 100 million people were estimated to have acquired the infection, contributing to a globally increasing burden of disease [3]. Although most infections are mild, dengue shock syndrome and dengue haemorrhagic fever are severe forms of infections and can be fatal [4,5]. The case-fatality rate can be as high as 20% in the absence of prompt diagnosis and lack of specific anti-viral drugs or vaccines [6,7], particularly in resource-limited settings. When an outbreak is particularly large, the influx of severe dengue cases can overwhelm the health system and prevent optimal care. Dengue also imposes an enormous societal and economic burden on many of the tropical countries where the disease is endemic [8]. An accurate prediction of the size of the outbreak and trends in disease incidence early enough can limit further transmission [5], and is likely to facilitate planning the allocation of healthcare resources to meet the demand during an outbreak.

Vector-borne pathogens characteristically demonstrate spatial heterogeneity—a result of spatial variation in vector habitat, climate patterns and subsequent human control actions [9–11]. The interplay of human, climate and mosquito dynamics give rise to a complex system that determines the pattern of dengue transmission, which in turn influences the potential for outbreak [12]. These relationships have been explored over the decades in the development of predictive models worldwide. Models vary widely in their purposes [13–15] and settings [16–21]. Many of these models excel at different tasks, however for a prediction model to be efficient, it requires a systematic, self-adaptive and generalizable framework capable of identifying weather and population susceptibility patterns across geographic regions. The scientific community has not yet agreed upon a model that provides the best prediction. The selection of predictors for the existing models is also quite heterogeneous. Some models rely solely on climate variables [16], some include vector characteristics [17,18] others use population

characteristics [19–21]. A wide range of statistical techniques are used with varying degrees of accuracy and robustness among the existing models [16–21].

Clarity in the documentation of the model development processes and model performance are essential for ensuring the robustness of the prediction [22], which is scarce as many of the existing models have not yet been systematically appraised. Given the disparate approaches, a focused synthesis and appraisal of the existing models, along with their building techniques and factor catchments, is required. Carefully establishing these details will provide the foundation for updating and developing robust models in future. This study aimed to systematically review all published literature that reported quantitative models to predict dengue outbreaks, revealing several shortcomings in the usage of real time primary predictive data and non-climatic predictors in the development of models, as well as inadequate reporting of techniques, model and performance measure validation.

## Methods

### Search strategy and selection criteria

This systematic review's aim, search strategy and study selection process were devised in accordance with the seven items in the Checklist for Critical Appraisal and Data Extraction for Systematic Reviews of Prediction Modelling Studies ('CHARMS') framework [23]. CHARMS framework is a systematic review tool, devised to facilitate and guide the methodological aspects the systematic review of prediction modelling studies, ranging from question development, appraisal of studies, and data extraction thereof. Detail of the CHARMS checklist can be found elsewhere [23]. The review followed the Preferred Reporting Items for Systematic Review and Meta-Analysis ('PRISMA') guidelines [24], and was registered in PROSPERO (CRD42018102100).

A literature search was conducted from inception until October 2022 using the electronic databases of Ovid MEDLINE, Embase, Scopus and Web of Science to obtain the information on the statistical models for predicting the number of dengue cases based on climatic factors. Google Scholar and the bibliography of included papers were also searched. The search strategies were developed under the guidance of an information specialist from Monash University Library. For the purposes of this study, dengue fever or dengue haemorrhagic fever or dengue shock syndrome were considered as a single entity "dengue". Search strategy included Medical Subject Headings ('MeSH') and keyword terms including "dengue", "severe dengue," "weather," "climate change," "model," "predict," and "forecast." The detailed search strategy and history are presented in S1 Table.

The review included studies focused on (1) prognostic prediction models which aim to review models predicting future events, (2) incidence of dengue fever or dengue haemorrhagic fever cases, (3) models to be used to predict the number of cases prior to an outbreaks, (4) models intended to inform public health divisions of future dengue outbreaks, (5) models with no restrictions on the time span of prediction and (6) prediction model development studies without external validation, or with external validation in independent data. Peer-reviewed original articles that presented a model and were available as full-text articles were considered eligible if they focused on predicting the number of dengue cases or an outbreak based on number of dengue incidence. Articles that focused on updating previously developed models were only included if they presented an updated version of the model. Articles which dealt exclusively with dengue in international travellers, or which only analyse the correlation between climate parameters and dengue cases without presenting a prediction model were excluded. Furthermore, articles which used models for predicting the population of dengue vectors (e.g., *Aedes aegypti* or *Aedes albopictus*) as well as articles which only offer susceptible-

infected-recovered modelling stochastic or transmission rates modelling were excluded. Articles which presented a model only dealing with spatial or temporal components of dengue risk were considered ineligible. Conference proceedings, book chapters, abstracts or letters were also excluded. Titles and abstracts of the retrieved articles were screened independently by two reviewers (RMI, MMA). Two review team members (LM, XYL) then retrieved the full text of those potentially eligible studies and independently assessed their eligibility. Disagreements were resolved by a third reviewer (MNK). A detailed study selection process is illustrated in the PRISMA flow diagram (Fig 1) [24].

## Data analysis

Based on the data extraction fields of the CHARMS framework [23], a standardised table was developed to extract data from the selected studies for assessment of quality and evidence synthesis. The data extraction table consists of eleven domains, each with a specific item, that extract data from the reports of the primary forecasting model. Key information extracted from the included articles were period and geographical region, sources of data, outcomes to be predicted, modelling covariates variables, sample size, statistical techniques, model performances, model evaluation, and key findings. Information regarding handling and/or reporting of missing data was also extracted. Each paper was independently reviewed by two reviewers (MMA, XYL) and discrepancies were resolved through discussion with each other or with a third reviewer (SP) where necessary.

Extracted data from the selected studies were summarised and the key information about the methodological characteristics of these models were tabulated. Descriptive statistics were generated based on model characteristics and comparative methodological features such as outcome types, target population, data sources and predictor selection techniques. All statistical analyses were performed using Stata (version 17.0).

## Results

The initial search yielded 6553 studies. After duplicates were removed, 3244 studies were screened for titles and abstracts. This led to 153 studies for full text review, and 64 that strictly met the inclusion criteria (Fig 1), 16 of these studies reported multiple models. A total of 99 models from 64 selected studies were identified. Characteristics of the models including, year, country and source of data used, predictors and outcome of the models, overall model development technique and model performance related variables are summarised in Table 1 [14,15,17–21,25–81].

Table 2 presents the sources of data used for modelling. Most of the models (90.7%) sourced dengue incidence data from surveillance and 44.3% used registry data, while 34.0% also used hospital or laboratory data. While most (94.7%) of the models used climate data from government agency reports, only around 22.1% of the models used data from the meteorological stations in real-time. Climate change data was also sourced mostly (77.8%) from government agency reports, only 11.1% used international environmental agency data and 22.2% used local environmental agency report. Half (50.0%) of the models used vector data from entomological surveillance and 25.0% used vector data from laboratory sources. Around 83.8% of the models were built based on the sample from general population, 16.2% used only urban samples. Around 46.9% of the models used monthly aggerate data, over a third (29.3%) used weekly aggregate data and 23.2% used daily aggregated data of the predictors. The majority (59.6%) of the models incorporated reporting time lag adjustment. Although 17.2% of the models addressed the missing data, 30.3% did not address the issue, while the majority (52.5%) did not specifically report the missing value. Around 80.8% of the models were intended to

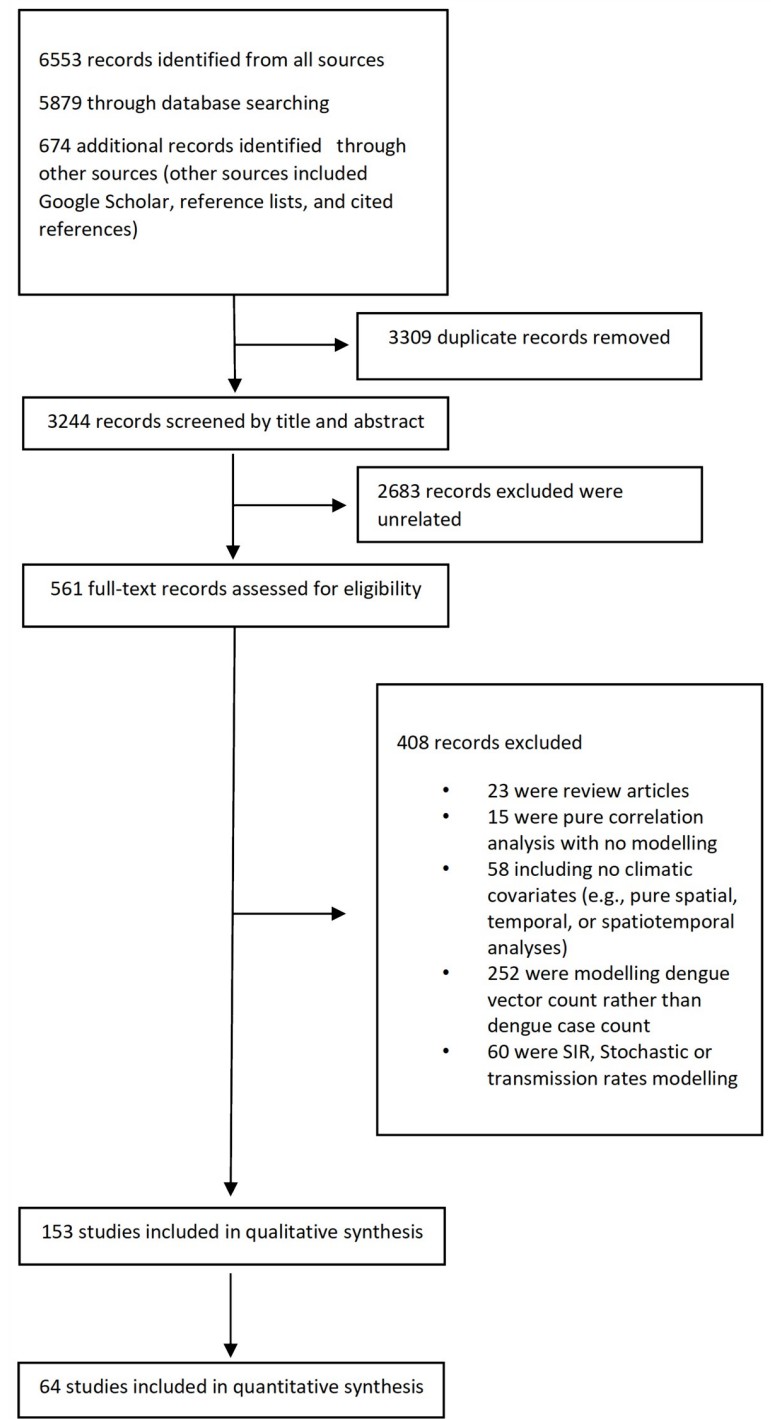

**Fig 1. PRISMA flow diagram illustrating study selection process.**

**Table 1. Characteristics of included predictive models.**

| Author and year | Country | Source of dengue data | Participant's recruitment | Outcome | Candidate predictors | Sample size | Missing data | Model development | Model techniques used | Model performance | Evaluation |
|---|---|---|---|---|---|---|---|---|---|---|---|
| Abualamah et al. (2021) [25] | Saudi Arabia | Surveillance, notification | Both | Case | Climate | Not reported | Not reported | All variable | Statistical model | BIC | Internal only |
| Aburas et al. (2010) [17] | Singapore | Notification | Urban | Case | Climate | 14209 | Not reported | All variable | Machine learning | Not reported | Internal only |
| Adde et al. (2016) [26] | France | Surveillance, laboratory | Urban | Outbreak | Climate, climate change | Not reported | Not reported | Pre-selection | Statistical model | AIC | Internal only |
| Anggraeni et al. (2017) [27] | Indonesia | Notification, registry | Both | Case | Climate | Not reported | Not reported | All variable | Statistical model | MAPE | Internal only |
| Bal et al. (2020) [28] | India | Notification | Urban | Case | Climate | Not reported | Not reported | All variable | Statistical model | AIC | Internal and external |
| Banu et al. (2015) [29] | Bangladesh | Notification | Both | Outbreak | Climate, climate change | Not reported | Not reported | Pre-selection | Statistical model | AIC, $R^2$ | Internal |
| Baquero et al. (2018) [30] | Brazil | Notification, hospital, registry | Both | Case | Climate | Not reported | Reported | Pre-selection, all variable | Statistical model, machine learning | RMSE | Internal only |
| Betanzos-Reyes et al. (2018) [20] | Mexico | Notification | Both | Case | Climate, vector | 2526 | Not reported | Pre-selection | Statistical model | $R^2$ | Not reported |
| Bett et al. (2019) [31] | Vietnam | Notification | Urban | Case | Climate, demography, environment | Not reported | Not reported | Pre-selection | Statistical model | Theil's coefficient of inequality | Internal only |
| Bouzid et al. (2014) [32] | Mexico | Laboratory | Urban | Case | Climate, demography | Not reported | Not reported | All variable | Statistical model | GCV | Internal only |
| Buczak et al. (2018) [33] | Peru | Surveillance, laboratory | Both | Case | Climate, environment | Not reported | Not reported | All variable | Machine learning | RMSE, MARE | Not reported |
| Buczak et al. (2018) [33] | Puerto Rico | Surveillance, laboratory | Both | Case | Climate, environment | Not reported | Not reported | All variable | Machine learning | RMSE, MARE | Not reported |
| Carvajal et al. (2018) [34] | Philippines | Notification | Urban | Case | Climate, climate change | Not reported | Reported | Pre-selection | Statistical model, machine learning | AIC, $R^2$ | Internal only |
| Chang et al. (2015) [21] | Taiwan | Surveillance, laboratory | Both | Case | Climate, vector | 8918 | Not reported | All variable | Statistical model | AIC, AUC, ROC | Internal only |
| Chen et al. (2022) [35] | China | Surveillance, registry | Both | Case | Climate | Not reported | Not reported | All variable | Statistical model | $R^2$, GCV | Internal only |
| Chen et al. (2022) [36] | China | Surveillance, registry | Both | Outbreak | Climate, vector | Not reported | Not reported | All variable | Machine learning | Retrospective forecasts | Internal only |
| Cheng et al. (2020) [37] | Taiwan | Notification, registry | Both | Case | Climate | Not reported | Not reported | All variable | Statistical model | AIC, BIC, MAE, RMSE | Internal only |
| Chuang et al. (2017) [38] | Taiwan | Notification | Urban | Outbreak | Climate, climate change | 71793 | Not reported | Pre-section | Statistical model | AIC | Internal only |
| Colón-González et al. (2013) [39] | Mexico | Surveillance | Urban | Case | Climate, demography, environment | Not reported | Not reported | All variable | Statistical model | Not reported | Internal only |

(*Continued*)

**Table 1.** (Continued)

| Author and year | Country | Source of dengue data | Participant's recruitment | Outcome | Candidate predictors | Sample size | Missing data | Model development | Model techniques used | Model performance | Evaluation |
|---|---|---|---|---|---|---|---|---|---|---|---|
| Depradine et al. (2004) [40] | Barbados | Notification, laboratory | Both | Case | Climate | Not reported | Not reported | All variable | Statistical model | $R^2$ | Not reported |
| Descloux et al. (2012) [41] | New Caledonia | Surveillance, laboratory, hospital | Both | Outbreak | Climate, climate change, vector | Not reported | Not reported | Pre-selection | Statistical model | AIC | Internal only |
| Dey et al. (2022) [42] | Bangladesh | Notification, registry | Both | Case | Climate | Not reported | Reported | All variable | Machine learning | MAE | Internal only |
| Dharmawardana et al. (2017) [43] | Sri Lanka | Notification | Both | Outbreak | Climate, environment | Not reported | Not reported | All variable | Machine learning | $R^2$ | Not reported |
| Earnest et al. (2012) [44] | Singapore | Notification, laboratory | Both | Outbreak | Climate, climate change | Not reported | Not reported | All variable | Statistical model | MAPE, $R^2$ | Not reported |
| Edussuriya et al. (2021) [45] | Sri Lanka | Notification | Both | Case | Climate, demography | Not reported | Reported | Pre-selection | Machine learning | RMSE | Not reported |
| Gharbi et al. (2011) [18] | France | Notification, laboratory | Both | Outbreak | Climate | Not reported | Not reported | All variable | Statistical model | AIC, RMSE | Not reported |
| Guo et al. (2017) [46] | China | Notification | Both | Case | Climate | Not reported | Not reported | All variable | Statistical model, machine learning | RMSE, $R^2$ | Internal only |
| Jain et al. (2019) [47] | Thailand | Notification | Both | Case | Climate | Not reported | Not reported | All variable | Statistical model | RMSE, $R^2$, AIC | Internal and external |
| Jayaraj et al. (2019) [48] | Malaysia | Notification, laboratory | Both | Both | Climate | Not reported | Not reported | Pre-selection | Statistical model | MAE, MSE | Internal only |
| Karim et al. (2012) [15] | Bangladesh | Notification | Urban | Case | Climate | 22705 | Not reported | All variable | Statistical model | ROC | Internal only |
| Lauer et al. (2018) [49] | Thailand | Notification, registry | Both | Case | Climate | Not reported | Not reported | Pre-selection | Statistical model | MAE | Internal only |
| Li et al. (2017) [50] | China | Notification | Both | Case | Climate | Not reported | Not reported | Pre-selection | Statistical model | Not reported | Internal only |
| Li et al. (2022) [51] | Brazil | Registry, hospital, laboratory | Both | Case | Climate, environment | Not reported | Not reported | Pre-selection | Machine learning | RMSE, MAE | Internal only |
| Liu et al. (2019) [52] | China | Notification, registry | Both | Case | Climate | Not reported | Not reported | All variable | Statistical model | RMSE, $R^2$, AIC | Internal only |
| Lowe et al. (2013) [53] | Brazil | Notification, laboratory | Urban | Case | Climate, climate change, demography | Not reported | Not reported | Pre-selection | Statistical model | AUC | External only |
| Luz et al. (2008) [54] | Brazil | Notification, registry | Both | Case | Climate | Not reported | Not reported | All variable | Statistical model | AIC, ACF, PACF | Internal only |
| McGough et al. (2021) [55] | Brazil | Notification, registry | Both | Case | Climate | Not reported | Not reported | Pre-selection | Machine learning | Out-of-sample forecast | External only |
| Mincham et al. (2019) [56] | China | Notification, laboratory | Both | Case | Climate, vector | Not reported | Not reported | All variable | Machine learning | sensitivity, specificity, PPV, NPV | External only |

(*Continued*)

**Table 1.** (Continued)

| Author and year | Country | Source of dengue data | Participant's recruitment | Outcome | Candidate predictors | Sample size | Missing data | Model development | Model techniques used | Model performance | Evaluation |
|---|---|---|---|---|---|---|---|---|---|---|---|
| Nakhapakorn et al. (2005) [57] | Thailand | Notification | Both | Case | Climate | Not reported | Not reported | All variable | Statistical model | Not reported | Internal only |
| Nan et al. (2018) [58] | China | Notification, registry | Both | Case | Climate | Not reported | Not reported | Pre-selection | Machine learning | RMSE, MAE, R² | Internal only |
| Nguyen et al. (2022) [59] | Vietnam | Surveillance, registry | Both | Case | Climate | Not reported | Reported | All variable | Machine learning | MAE | Internal only |
| Nuraini et al. (2021) [60] | Indonesia | Notification | Both | Case | Climate | Not reported | Not reported | All variable | Statistical model | Least square method | Not reported |
| Olmoguez et al. (2019) [61] | Philippines | Notification | Both | Case | Climate | Not reported | Not reported | All variable | Statistical model, machine learning | R², MAPE | Not reported |
| Pham et al. (2018) [62] | Malaysia | Notification, registry | Both | Case | Climate | Not reported | Not reported | Pre-selection | Statistical model, machine learning | RMSE, MAE, Scatter plot | Not reported |
| Pham et al. (2020) [63] | Vietnam | Notification | Both | Case | Climate | Not reported | Not reported | All variable | Statistical model | Retrospective forecasts | Internal only |
| Phung et al. (2015) [64] | Vietnam | Notification, laboratory | Both | Case | Climate | 13509 | Not reported | All variable | Statistical model | AIC, BIC, ROC | Internal only |
| Phung et al. (2016) [14] | Vietnam | Surveillance/ notification | Both | Case | Climate | Not reported | Not reported | All variable | Statistical model | ROC | Internal only |
| Pineda et al. (2019) [65] | Philippines | Notification | Both | Case | Climate | Not reported | Not reported | All variable | Statistical model | Retrospective forecasts | Internal only |
| Pinto et al. (2011) [66] | Singapore | Notification | Urban | Case | Climate | Not reported | Not reported | All variable | Statistical model | Not reported | Internal only |
| Puengpreeda et al. (2020) [67] | Thailand | Notification, registry | Both | Case | Climate | Not reported | Addressed | All variable | Machine learning | MSE, MAE, MAPE, R² | Internal only |
| Qureshi et al. (2017) [19] | Pakistan | Notification | Urban | Outbreak | Climate, vector | Not reported | Not reported | All variable | Machine learning | Not reported | Not reported |
| Ramadona et al. (2016) [68] | Indonesia | Notification | Both | Case | Climate | 7171 | Not reported | All variable | Statistical model | AIC, RMSE | External only |
| Roster et al. (2021) [69] | Brazil | Notification | Both | Outbreak | Climate | Not reported | Not reported | All variable | Machine learning | MAE, RMSE | Internal only |
| Salim et al. (2021) [70] | Malaysia | Notification, registry, laboratory | Both | Outbreak | Climate, vector | Not reported | Not reported | All variable | Statistical model, machine learning | ROC | Internal only |
| Shi et al. (2016) [71] | Singapore | Notification | Urban | Case | Climate, vector | Not reported | Not reported | All variable | Statistical model | MAPE | Internal only |
| Siriyasatien et al. (2016) [72] | Thailand | Notification | Both | Case | Climate, vector, demography | Not reported | Not reported | All variable | Statistical model | AIC, BIC | Internal only |
| Withanage et al. (2018) [73] | Sri Lanka | Notification | Both | Case | Climate | 56843 | Not reported | All variable | Statistical model | MAPE, RMSE, MAE, PSS | Internal only |

(*Continued*)

**Table 1.** (Continued)

| Author and year | Country | Source of dengue data | Participant's recruitment | Outcome | Candidate predictors | Sample size | Missing data | Model development | Model techniques used | Model performance | Evaluation |
|---|---|---|---|---|---|---|---|---|---|---|---|
| Xu et al. (2020) [74] | China | Notification | Both | Case | Climate | Not reported | Not reported | Pre-selection | Machine learning | Not reported | Internal only |
| Yuan et al. (2019) [75] | Taiwan | Notification, registry | Both | Case | Climate | Not reported | Not reported | Pre-selection | Statistical model | AIC | Internal only |
| Yuan et al. (2020) [76] | China | Notification, laboratory | Both | Case | Climate | Not reported | Not reported | Pre-selection | Statistical model | MSE, NMSE | Internal only |
| Zafra, B (2020) [77] | Philippines | Notification, registry | Both | Case | Climate | Not reported | Reported | All variable | Machine learning | Not reported | Internal only |
| Zambrano et al. (2012) [78] | Honduras | Laboratory | Both | Case | Climate, climate change | 3353 | Not reported | All variable | Statistical model | Not reported | Not reported |
| Zhang et al. (2016) [79] | China | Notification, laboratory | Both | Case | Climate | 38150 | Not reported | All variable | Statistical model | AIC, ROC | Internal only |
| Zhao et al. (2020) [80] | Colombia | Notification | Both | Case | Climate, climate change, demography, environment | Not reported | Not reported | All variable | Machine learning | MAE, RMAE | Internal only |
| Zhu et al. (2019) [81] | China | Notification, laboratory | Both | Case | Climate | Not reported | Not reported | Pre-selection | Statistical model | $R^2$ | Not reported |

AIC = Akaike information criterion, ACF = autocorrelation function, AUC = area under curve, BIC = Bayesian information criterion, GCV = generalized cross-validation score, MAE = mean absolute error, MAPE = mean absolute percentage error, MARE = mean absolute relative error, MSE = mean squared error, PACF = partial autocorrelation function, NMSE = normalised mean squared error, PSS = Pierce skill score, RMAE = relative mean absolute error, ROC = receiver operating characteristic, RMSE = root mean squared error, PPV = Positive predictive value, NPV = Negative predictive value

**Table 2. Source of data used for modelling.**

| Sources of model data | N (%) * |
|---|---|
| *Dengue data source (n = 99)* | |
| Surveillance/notification | 88(90.7) |
| Disease registry | 43(44.3) |
| Hospital/laboratory | 33(34.0) |
| *Climate data source (n = 99)* | |
| Government agency report | 90 (94.7) |
| Meteorology station | 21 (22.1) |
| Research Institute/centre | 4 (4.2) |
| *Climate change data source (n = 18)* | |
| Government agency report | 14 (77.8) |
| International environmental agency | 2 (11.1) |
| Local environmental agency report | 4 (22.2) |
| *Vector data source (n = 8)* | |
| Surveillance/monitoring data | 4 (50.0) |
| Laboratory data | 2 (25.0) |
| Government agency report | 3 (37.5) |
| *Population source (n = 99)* | |
| General population | 83 (83.8) |
| Metropolitan | 16 (16.2) |
| *Data aggregation unit (n = 98)* | |
| Daily aggregate | 23 (23.2) |
| Weekly aggregate | 29 (29.3) |
| Monthly aggregate | 46 (46.9) |
| *Lag time adjusted in model (n = 99)* | |
| No | 40 (40.4) |
| Yes | 59 (59.6) |
| *Treatment of missing data (n = 99)* | |
| Yes | 17 (17.2) |
| No | 30(30.3) |
| Not reported | 52 (52.5) |
| *Prediction outcome (n = 99)* | |
| Dengue case count | 80 (80.8) |
| Dengue outbreak | 19 (19.2) |

* The percentages may not add up to 100, due to multiple responses

predict the number of dengue cases and 19.2% focused on predicting dengue outbreaks, based on predetermined case number threshold.

Table 3 summarises the statistical methods adopted by the prediction models. Modelling techniques were broadly categorised under two genres, statistical models (60.6%) and machine learning (39.4%). The statistical models were broadly comprised of linear regression models (18.3%), time series/autoregressive models (26.7%), Poisson regression models (18.3%) and generalized additive models (16.7%). Neural networks models (23.1%), random forest models (15.4%), and ensemble models (10.3%) were types of machine learning models used.

All theoretically plausible predictors were considered as candidate predictors in 71.7% of models and pre-selection of predictors based on unadjusted association with the model outcome was considered in 28.3% of models. Reporting of essential modelling techniques was

**Table 3. Statistical methods used among models (n = 99).**

| Statistical methods | N (%) * |
|---|---|
| *Model building technique* | |
| *Statistical models (n = 60, 60.6%)* | |
| Linear regression model | 11 (18.3) |
| Non-linear regression model | 5 (8.3) |
| Time series/autoregressive model | 16 (26.7) |
| Poisson regression model | 11 (18.3) |
| Generalized linear model (GLM) | 2 (3.3) |
| Generalized additive model (GAM) | 10 (16.7) |
| Others | 5 (8.3) |
| *Machine learning (n = 39, 39.4%)* | |
| Random forest | 6 (15.4) |
| Neural network | 9 (23.1) |
| Boosting algorithm | 5 (12.8) |
| Support vector algorithm (SVA) | 4 (10.3) |
| Ensemble models | 4 (10.3) |
| Classification and regression tree (CART) | 2 (5.1) |
| Long short-term memory (LSTM) | 4 (10.3) |
| Others | 5 (12.8) |
| *Predictor selection for model* (n = 99) | |
| All theoretically plausible predictors | 71 (71.7) |
| Pre-selection (unadjusted association) | 28 (28.3) |
| Reporting model parameter (n = 99) | |
| Model performance | 84 (84.8) |
| Model calibration | 58 (58.6) |
| Model discrimination | 47 (47.5) |
| Model validation (n = 99) | |
| External and internal validation | 5 (5.2) |
| Internal validation | 75 (75.8) |
| No validation | 20 (20.2) |
| Model validation techniques (n = 79) | |
| Split sample validation | 16 (20.3) |
| Cross validation | 32 (40.5) |
| Retrospective validation | 5 (6.3) |
| Out of sample validation | 3 (3.8) |
| Performance metrices only | 23 (29.1) |
| Performance metrices reported | |
| MAE (Mean Absolute Error) | 7 (7.1) |
| RMSE (Root Mean Squared Error) | 11 (11.1) |
| AIC/BIC (Akaike / Bayesian Information Criterion) | 1 (1.0) |
| ROC (Receiver Operating Characteristic) | 5 (5.1) |
| MAPE (Mean Absolute Percentage Error) | 5 (5.1) |
| GCV (Generalized Cross Validation score) | 2 (2.0) |
| MSE (Mean Squared Error) | 7 (7.3) |

* The percentages may not add up to 100 as studies used multiple methods

heterogeneous– 84.8% of models reported model performance, 58.6% reported model calibration and 47.5% reported model discrimination. Among the performance metrics, Root Mean Squared Error ('RMSE') (11.1%), Mean Squared Error ('MSE') (7.3%), Mean Absolute Percentage Error ('MAPE') (5.1%), and Receiver Operating Characteristic ('ROC') (5.1%) were notable. Of these models, most (75.8%) reported the internal validation alone, only 5.2% reported both internal and external validation and 20.2% reported no validation at all. The validation techniques included: split sample validation (development and validation) (20.3%), cross validation, which involves resampling of the derivation sample (40.5%) and performance metrics (29.1%).

Table 4 presents the factors used for prediction models. All of the models included in the review used climate predictors in their model. Among the climate predictors: humidity (77.4%), temperature (95.2%) and rainfall (81.0%), were used in most models. Windspeed and direction (27.4%), precipitation (15.5%) and sunshine (10.7%) were among other notable climate factors. Considering the similarity of the description of factors, climate change and environmental factors were collapsed in to one category under climate change. Overall, 18.2% of the models used climate change and/or environmental predictors. El Nino-Southern Oscillation ('ENSO'), Southern Oscillation Index ('SOI'), Oceanic Nino Index ('ONI'), hydric balance and vegetation index were among the key climate change predictors. Vegetation Index and enhanced vegetation index were among the key environmental factors reported.

Vector-related predictors were included in 8.1% of models, and the key vector related predictors were container index, Breteau index, adult productivity index, breeding percentage and mosquito infection rate. Demographic predictors were included in 8.1% of models, and key demographic predictors were, population size, population density, area under the urban settlement, access to piped water, education coverage and GDP per capita (Table 4).

The combination of the predictors used in the model are depicted in Fig 2. While majority (70.7%) of the models were built solely on climate predictors, none of the models used the

**Table 4. Factors that appeared as predictors in the prediction models.**

| Climate factors (100.0%) | Climate change and environmental factors (18.2%) | Entomological (Vector) factors (8.1%) | Demographic factors (8.1%) |
|---|---|---|---|
| *Temperature (95.2%)* Minimum temperature Mean temperature Maximum temperature *Rainfall (81.0%)* Average rainfall Accumulated rainfall Number of rainy days *Humidity (77.4%)* Relative humidity Absolute humidity *Sunshine (10.7%)* Sunshine duration Insolation *Windspeed & direction (27.4%)* *Precipitation (15.5%)* *Evaporation (8.3%)* *Atmospheric pressure (2.4%)* | *El Nino-Southern Oscillation (ENSO)* *Southern Oscillation Index (SOI)* *Oceanic Nino Index (ONI)* *Gini Index* *Potential evapotranspiration* *Azores high sea-level pressure* *Dipole mode index* *Hydric balance* *Vegetation Index* *Enhanced vegetation index* *Equatorial Pacific Ocean surface temperature* | *Ades albopictus count* *Container index* *Ades aegypti index* *Breteau index* *Adult productivity index* *Weekly egg count in ovitrap* *Breeding percentage* *Mosquito infection rate* *Minimum infection rate* *Per man hour density (PMHD)* | *Population size* *Population density* *Access to piped water* *Education coverage* *GDP per capita* *Area under urban settlement* |

\* Figures in the parenthesis denotes the percentage of models included the predictor in the model and percentages may not add up to 100 as models used multiple categories of predictors in combination

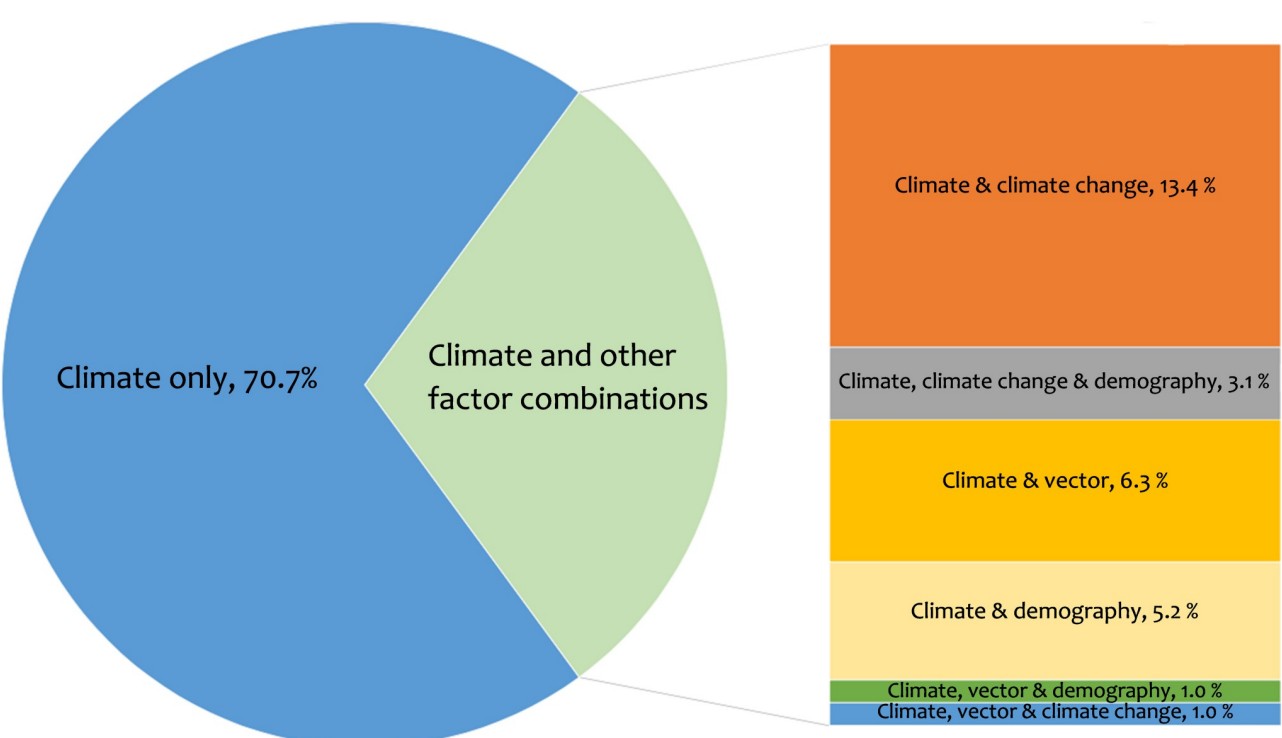

**Fig 2. Combinations of predictors used in the prediction models.**

combination of all four (climate, climate change, vector and demography) categories of predictors. The combination of climate, climate change and demographic predictors was used in 3.1% of the models and the combination of climate and climate change predictors were used in 13.4% models. Among other notable combinations were, climate and vector predictors (6.3%) and climate and demographic predictor (5.2%).

## Discussion

This systematic review evaluated 99 dengue outbreak prediction models from 64 studies, predicting the number of dengue cases or outbreaks from a variety of settings and populations. Our review identified, three major area of inadequacy in the current modelling practices. Firstly, use of secondary predictor data—acquired from reports—were quite prevalent among models. Secondly, as data for other non-climate variables were not included in the majority of the models, they failed to capture a holistic view of dengue development in the prediction process. Lastly, inadequacy in the reporting of methodology, model validation and performance measures were quite prevalent in the existing prediction models. One positive aspect seen in the current modelling practice is the shift toward robust modelling technique, such as use of machine learning algorithm and autoregressive time series techniques.

While effective treatments and prevention measures are still being developed, an early warning system for an epidemic has the potential to reduce the toll of severe disease on the health system and population [82]. Developing a clearer understanding of the factors affecting dengue transmission is an important step towards mitigating the impact of the disease on health systems and on communities at large. Early prediction of dengue incidences or alerts regarding impending outbreak may contribute to the health system preparedness through

effective resource mobilization and creating public awareness. Such predictions also have policy implications, as epidemiological evidence generated through modelling feeds the policy making process and facilitates the prioritization of interventions, such as vector control and environmental modification particularly in regard to climate change [83]. Considering dengue is a mosquito-borne disease, the majority of outbreak prediction models focus on climate dependency of mosquito breeding and dengue transmission [4–7]. While many models have been successful in predicting relative cases of dengue in real settings, incorrect prediction results have been observed in several included studies. For example, a model by Adde et al. [26] was unable to forecast a dengue outbreak in 2001–2005 with the use of climate data from 1991–2000. One of the potential reasons for their inaccurate prediction is the geo-spatial variation of climate and environment within regions. In their study, the decision to include vastly heterogenous geographical areas led to variation in model prediction, which—be due to exclusion of non-climatic factors—may be the explanation of the poor performance in many of the earlier prediction models could. An increasing number of later models appears to incorporate a wide range of vector parameters as well as demographic parameters. Chang et al. pointed out that, entomological (vector) factor combined with other meteorological (climate) factors, have better prediction performance, and their prediction accuracy is often higher than that of climate predictors alone [21].

For dengue incidence data, the majority of the models relied on reports from government organizations based on notifiable data. Notification involves passive surveillance, where there is potential for systematic underreporting along with varying time lag. Modelling with data from active surveillance or real-time study may minimize such limitations. A considerable number of models did not consider the time lag affecting the prediction, which may be responsible for possible delays in weather affecting mosquito vectors and subsequently viruses. Due to the nature of dengue disease dynamics, failure to address time lag in model development is likely to affect prediction accuracy. Critical points in the natural history of disease timeline those may generate time lag may start with mosquito development, and subsequently also during acquisition and amplification virus in mosquitoes, mosquito host behaviour (i.e. biting and feeding pattern) and the incubation period of the virus in the human body [12,48]. Some studies have found a positive correlation between climate variables with time-lags at several points in the natural history of disease timeline [48,53]. Therefore, the adjustment for the time lags while predicting dengue is indispensable, especially when meteorological data is used [12].

The majority of included models were built on conventional regression techniques. According to recent literature, the time series technique is particularly considered effective in predicting the highly auto-correlated nature of dengue infection [84,85]. Machine learning techniques are employed in around 40% of the included models, and is particularly prevalent among the recently developed models. Batista et al. confirmed superiority with ML techniques demonstrating a lower error rate compared to the conventional statistics-based model in predicting dengue cases. In the age of big data, this technique can leverage data availability and in addition to being non-parametric in nature, can also provide some leeway in terms of strict assumption [86]. Random forest, neural networks, gradient boosting and support vector algorithms are notable subsets of machine learning algorithms, which have made significant contributions to several areas of public health, particularly in the forecasting of infectious diseases like malaria [87] and COVID-19 [88], and may have similar utility for making dengue outbreak predictions. Although machine learning in gaining popularity, future modelling in this area may benefit from using mechanistic models [89]. This modelling technique have played an essential role in shaping public health policy over the past decades [90]. Mechanistic models have the potential to provide additional insight regarding precise dynamics of the transmission and infection of

dengue. As these models highlight underlying processes that drive the patterns. These models can particularly aid in the prediction through incorporating the observed trajectory of vectors.

In the modelling process, generating an algorithm or equation is only part of the process. It is not complete unless its performance has been assessed considering discrimination [91] and calibration [92], both internally and using the population outside of what it is developed from, respectively. Among the existing models examined, reporting of the discrimination and calibration is very low. Without knowledge of model performance through validation in both source populations and populations other than where it was developed, objective evaluation of models is difficult [93]. Predictive models can be of great value only if there is certainty of its accuracy, that is, how precisely the model can predict an outcome in real world [94]. In the majority of the published models, real-world validation has not been performed or reported. Generally, models are unlikely to predict as well in real-world samples as it would in the derivation sample; this validity shrinkage can often be quite substantial. Hence, future models should report a mechanism of estimating and reporting potential validity shrinkage as well as predictive validity in real world data [95, 96].

In a substantial proportion of the models that reported validation, the original dataset was randomly split into the development and validation subset. Although this approach is widely used in many model validation settings, there are some setbacks when using smaller operational sample sizes, as split-sample analyses give overly pessimistic estimates of model performance and are accompanied by large variability [97]. Bootstrapping is generally considered to be the preferred internal validation method in predictive models [98, 99]. Interestingly, bootstrapping was not used in any of the models in included studies, instead cross-validation technique was adopted in most of them. External validation, on the other hand, was used only in very few included studies. This is despite the fact that external validation is considered pivotal to model development and a key indicator of model performance through highlighting applicability to participants, centres, regions or environments [23]. The external validation is particularly essential for model redevelopment, where the original model is adjusted, updated, or recalibrated based on validation data to improve performance [100]. This update may include adjusting the baseline risk (interception or hazard) of the original model, adjusting the weight or regression coefficient of the predictor, adding new predictors, or removing existing predictors from the model.

This review has a number of strengths–specifically, the use of the CHARMS checklist [23], designed for the assessment of the applicability of the prediction models. In addition, inclusion and exclusion criteria were strictly followed, and database searches were conducted by an expert librarian. However, there are a few limitations of the review–the models in this review are not explicitly rated based on quality or performance due to the lack of accepted criteria for rating the quality of forecasting models. In addition, although calibration was reported in several studies, calibration measures lack clarification, which may impact the overall evaluation of the model performance. The model performance could not be compared across methodological approaches in quantitative synthesis because of a lack of model performance data, and those that did provide data are mostly generated from internal validation data which may result in overfitting.

## Conclusion

In conclusion, failure to use of real time primary predictor data, failing to incorporate non-climatic parameters as predictor and insufficient reporting of model development techniques, model validation and performance measure were the major inadequacies identified in the current modelling practice. The paradigm shift towards robust modelling techniques, such as the use of machine learning algorithms and autoregressive time series, is a significant positive

trend in contemporary model practices. The findings of this review have the potential to lay the groundwork for improved modelling practices in the future. These findings will contribute to robust modelling in different settings and populations and have important implications for the planning and decision-making process for early dengue intervention and prevention.

## Supporting information

**S1 Table. Search strategy for OVID Medline, as performed in October 2022.**
(DOCX)

**S2 Table. PRISMA checklist.**
(DOCX)

## Author Contributions

**Conceptualization:** Dragan Ilic, Saif U. Munshi, Md Nazmul Karim.

**Data curation:** Xing Yu Leung, Rakibul M. Islam, Mohammadmehdi Adhami, Lara McDonald, Shanika Palawaththa.

**Formal analysis:** Md Nazmul Karim.

**Methodology:** Md Nazmul Karim.

**Project administration:** Xing Yu Leung, Rakibul M. Islam, Mohammadmehdi Adhami, Lara McDonald, Shanika Palawaththa.

**Supervision:** Dragan Ilic, Basia Diug.

**Visualization:** Shanika Palawaththa, Basia Diug.

**Writing – original draft:** Xing Yu Leung, Saif U. Munshi, Md Nazmul Karim.

**Writing – review & editing:** Xing Yu Leung, Rakibul M. Islam, Mohammadmehdi Adhami, Dragan Ilic, Lara McDonald, Shanika Palawaththa, Basia Diug, Saif U. Munshi, Md Nazmul Karim.

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
