## [Decision Letter · Decision Letter 0]

13 Sep 2022

Dear Dr. Karim,

Thank you very much for submitting your manuscript "A systematic review of dengue outbreak prediction models: current scenario and future directions" for consideration at PLOS Neglected Tropical Diseases. As with all papers reviewed by the journal, your manuscript was reviewed by members of the editorial board and by several independent reviewers. In light of the reviews (below this email), we would like to invite the resubmission of a significantly-revised version that takes into account the reviewers' comments. 

We cannot make any decision about publication until we have seen the revised manuscript and your response to the reviewers' comments. Your revised manuscript is also likely to be sent to reviewers for further evaluation.

Sincerely,

Husain Poonawala

Academic Editor

Stuart Blacksell

Section Editor

Reviewer's Responses to Questions

**Key Review Criteria Required for Acceptance?**

**Methods**

-Are the objectives of the study clearly articulated with a clear testable hypothesis stated?

-Is the study design appropriate to address the stated objectives?

-Is the population clearly described and appropriate for the hypothesis being tested?

-Is the sample size sufficient to ensure adequate power to address the hypothesis being tested?

-Were correct statistical analysis used to support conclusions?

-Are there concerns about ethical or regulatory requirements being met?

Reviewer #1: The study selection process in this review paper is appropriate.

Reviewer #2: The objectives of this study were clearly articulated. The study design and methods applied were appropriate to address the stated objectives. The sample size was sufficient to ensure adequate power to address the hypothesis being tested. No concerns about the ethical aspect.

Please consider including recently published papers rather included only papers published by July 2021.

Should explain CHARMS= Checklist for critical Appraisal and data extraction for systematic Reviews of prediction Modelling Studies.

Reviewer #3: (No Response)

**Results**

-Does the analysis presented match the analysis plan?

-Are the results clearly and completely presented?

-Are the figures (Tables, Images) of sufficient quality for clarity?

Reviewer #1: There are a few points missing in the results and discussion. Please see my detailed questions below.

Reviewer #2: The analysis presented matches the analysis plan. The results were clear and completed presented. The tables were clear, but the titles of the tables should be located on top of the tables.

Page 13, line 150: entomological data  should specify this was monthly or quarterly data.

Table 2. There was an error at "Climate data source"

Page 16, lines 180-184: how about environmental factors? Why these factors were not included in the analysis?

Figure 2. Why environmental factors were not included?

Page 20, lines 255-258: Please note that these models were used in recent publications, so you may consider including recent publications in this review.

Reviewer #3: (No Response)

**Conclusions**

-Are the conclusions supported by the data presented?

-Are the limitations of analysis clearly described?

-Do the authors discuss how these data can be helpful to advance our understanding of the topic under study?

-Is public health relevance addressed?

Reviewer #1: Conclusions are supported by the data.

Reviewer #2: The conclusions were supported by the data presented. The limitations of the analysis were described in the discussion. The authors already discussed how these results can be helpful to advance our understanding of the models used in forecasting dengue fever. 

Page 22, lines 296-297: Please note that deep learning models were used in recent publications. Please include publications in 2022, e.g. “Deep learning models for forecasting dengue fever based on climate data in Vietnam”, to ensure the results of this review are up to date.

Reviewer #3: (No Response)

**Editorial and Data Presentation Modifications?**

Reviewer #1: Accept.

Reviewer #2: Since this study was a systematic review of dengue outbreak prediction models: current scenario and future directions, the papers included in the review should be up to date. However, this review only included papers published by July 2021 while from August 2021 until July 2022, there were some important relevant papers being published that applied deep learning models. So, if the authors can include some more recently published papers, then the quality of the manuscript would be greatly improved.

Reviewer #3: (No Response)

**Summary and General Comments**

Reviewer #1: This is a comprehensive review of existing models to generate predictions for dengue. Commonly used data sources, predictors, and statistical methods were reported. However, I think there are a few key topics missing in the current manuscript.

1. Prediction can connate different tasks. It could be predicting the number of infections in the next few weeks/months, or the trend of disease over years/decades under different climate scenarios. I suggest the authors clearly define what prediction represents in this review.

2. There is a lack of discussion on the use of process-based or mechanistic models to generate predictions. I doubt that there were no studies using these types of models in the literature.

3. I suggest the authors to include a discussion on the utility of these model predictions. Were these predictions used to inform control policies? Or increase the public awareness?

4. Real-world validation of predictive models are essential to validate the predictive skills in application. In the US, real-time dengue forecast challenge was organized by the CDC to systematically compare different methods. See Johansson et al. An open challenge to advance probabilistic forecasting for dengue epidemics. PNAS. 116:24268-74, 2019. I think it is important to discuss future efforts to evaluate predictive models in real world settings.

5. It would be great to present more details on the predictions. Did the included studies generate point predictions or probabilistic predictions? What are the forecast horizons? What metrics were used to evaluate performance?

Reviewer #2: Overall, this manuscript was quite well written and presented an important topic. It would be better if authors consider including more recently published papers. Also, consider some comments in the PDF file attached to improve the quality of the manuscript.

Reviewer #3: (No Response)

PLOS authors have the option to publish the peer review history of their article (what does this mean?). If published, this will include your full peer review and any attached files.

Reviewer #1: No

Reviewer #2: No

Reviewer #3: Yes: Ignacio Sánchez-Gendriz
---

## [Decision Letter · Decision Letter 1]

29 Jan 2023

Dear Dr. Karim,

We are pleased to inform you that your manuscript 'A systematic review of dengue outbreak prediction models: current scenario and future directions' has been provisionally accepted for publication in PLOS Neglected Tropical Diseases.

Best regards,

Husain Poonawala

Academic Editor

Stuart Blacksell

Section Editor

Reviewer's Responses to Questions

**Key Review Criteria Required for Acceptance?**

**Methods**

-Are the objectives of the study clearly articulated with a clear testable hypothesis stated?

-Is the study design appropriate to address the stated objectives?

-Is the population clearly described and appropriate for the hypothesis being tested?

-Is the sample size sufficient to ensure adequate power to address the hypothesis being tested?

-Were correct statistical analysis used to support conclusions?

-Are there concerns about ethical or regulatory requirements being met?

Reviewer #1: Accept.

Reviewer #2: The authors have adequately addressed comments and questions raised in the 1s revision round. The objectives of the study were clearly defined. The search strategy, selection criteria and data analysis were appropriate.

Reviewer #3: - The objectives of the study are well explained and addressed.

- The authors include sufficient literature to support their review.

**Results**

-Does the analysis presented match the analysis plan?

-Are the results clearly and completely presented?

-Are the figures (Tables, Images) of sufficient quality for clarity?

Reviewer #1: Accept.

Reviewer #2: In the abstract and result sections, please check these data "Climate factors were used in combination with climate change factors (10.3%), both climate change and demographic factors (10.3%), vector factors (5.1%), and demographic factors (5.1%)" as I see these were exactly the same with those in the previous version of the manuscript, while you have recently added 10 following studies in the analysis:

Abualamah et al (2021), Baquero et al (2018), Chen et al (2022), Chen et al (2022), Cheng et al (2020), Dey et al (2022), Nguyen et al (2022), Pham et al (2020), Pineda et al (2019), Yuan et al (2019)

Reviewer #3: - The Results are clearly presented;

- The analyzes presented are compatible with the established objectives;

**Conclusions**

-Are the conclusions supported by the data presented?

-Are the limitations of analysis clearly described?

-Do the authors discuss how these data can be helpful to advance our understanding of the topic under study?

-Is public health relevance addressed?

Reviewer #1: Accept.

Reviewer #2: In the discussion and conclusion, the authors stressed that "the majority of the models ignored non-climatic variables". I think using the word "ignored" may be inappropriate in most cases, simply because data on other non-climatic variables were not available for analysis.

Reviewer #3: - The contribution of the study to public health is clearly addressed;

**Editorial and Data Presentation Modifications?**

Reviewer #1: Accept.

Reviewer #2: (No Response)

Reviewer #3: (No Response)

**Summary and General Comments**

Reviewer #1: The authors have addressed my questions.

Reviewer #2: In the introduction section, the authors cited data published 20 years ago. For example: "Every year around 390 million people acquire the infection, of which around 96 million develop clinical manifestations". So, please revise this section and cite the most updated data from recent peer-reviewed publications.

Reviewer #3: (No Response)

PLOS authors have the option to publish the peer review history of their article (what does this mean?). If published, this will include your full peer review and any attached files.

Reviewer #1: No

Reviewer #2: No

Reviewer #3: No

---

## [Editor Report · Acceptance letter]

8 Feb 2023

Dear Dr. Karim,

We are delighted to inform you that your manuscript, "A systematic review of dengue outbreak prediction models: current scenario and future directions," has been formally accepted for publication in PLOS Neglected Tropical Diseases.

Best regards,

Shaden Kamhawi

co-Editor-in-Chief

Paul Brindley

co-Editor-in-Chief
